# Vibroacoustic Study in the Neonatal Ward

**DOI:** 10.3390/healthcare10071180

**Published:** 2022-06-24

**Authors:** Jose Miguel Sequí-Canet, Romina del Rey-Tormos, Jesús Alba-Fernández, Gema González-Mazarías

**Affiliations:** 1Departamento de Pediatría, Hospital Universitario Francesc de Borja, 46702 Gandia, Spain; 2Centro de Tecnologías Físicas, Universitat Politècnica de València, 46020 Valencia, Spain; roderey@fis.upv.es (R.d.R.-T.); jesalba@fis.upv.es (J.A.-F.); gegonmaz@gmail.com (G.G.-M.)

**Keywords:** vibrations, neonates, environmental pollution

## Abstract

Neonatal wards are often subject to excessive noise pollution. Noise pollution encompasses two concepts, noise and vibration; their main difference being that a noise is heard and a vibration is felt in the body. The latter is what can be transmitted within the incubators of neonatal inpatients. This fact needs to be explored in depth. This work shows the results of the diagnosis of vibrations transmitted within the incubators that could affect neonates admitted to the neonatal unit of the Francesc de Borja University Hospital (Gandía, Spain). For this purpose, the vibrations reaching the neonate head resting area were recorded, taking into account different points, scenarios, days, and time slots. It could be observed that due to the incubator’s motor position, the levels obtained in some scenarios measured in this study exceeded the regulation-specified levels. The conclusion is that the greatest influence on vibrations is the incubator’s own motor, with other possible sources of vibrations, such as the room’s air conditioning, having less influence. Further studies are needed to determine whether this vibration is harmful or beneficial to the neonate.

## 1. Introduction

The concern for care quality in neonatology has been a long-standing concern of many pediatric associations [1]. The study of noise levels in neonatal intensive care units (NICU) is a topic of concern to the pediatric society and has therefore been extensively studied [2,3,4,5,6,7,8,9,10,11,12,13,14,15,16]. These studies show that there is an excessive level of noise in many cases. In these studies, the aim is to find not only the diagnosis of noise in the room but also different solutions for this excessive noise, which according to the World Health Organization (WHO), in some cases, can be considered noise pollution. There are also some studies on noise levels in pediatric neonatal units (NU) [17,18], which are much more numerous than NICUs. All the works referenced above are examples of many studies on what is known as noise. In reality, however, what is being evaluated—from an acoustic engineering standpoint—is the sound pressure level or airborne noise.

Environmental Commissioners from EU member states [19] define noise pollution as “the vibration or noise environment, whatever their acoustic source, which causes annoyance, risk or damage to people, their activities or property of any kind, or which causes significant effects on the environment.” With this definition, it is clear that noise pollution encompasses these two concepts, noise and vibration. The main difference is that a noise is heard and a vibration is felt in the body. The frequency spectra of each are well-defined. We can define airborne noise (which is heard) as the propagation of sound traveling from 20 Hz to 20,000 Hz. Vibration can only be produced by solids. Its perception is studied in the frequency range from 20 Hz to 80 Hz, as this range has the greatest effect on health [20].

It is almost impossible for anyone to avoid exposure to vibration in today’s world. The harmful effects of vibrations and their effect on humans have been researched and documented worldwide. Exposure can lead to large variations between subjects with regard to biological effects. Very often, it is the lumbar spine and the connected nervous system which can be affected by exposure. Other studies have highlighted the neck-shoulder, gastrointestinal system, female reproductive organs, peripheral veins, and the cochleo-vestibular system [21,22,23]. The effects are complex and depend at least on the amplitude, direction, frequency, duration of the vibration, and to which part of the body it is directed.

Most of these studies deal with workers exposed to long periods of vibration both from the machinery at work and from the transport used, and all have concluded that due to exposure, they can experience loss of balance, fatigue, reduced comfort, loss of concentration and even health risks, with the possibility of damaging some organs when exposed to certain frequencies or amplitudes. However, when it comes to the effects of vibrations on a neonate, there is still a great deal of research to be completed.

As aforementioned, some of the studies have focused on the study of noise in NICUs or NUs, but all these works focus on the assessment of airborne noise, with the evaluation of sound propagation between the frequencies of 100 Hz up to 5000 Hz. If we wish to know the vibrations found in pediatric environments, there is very little information available, as this type of study is focused on the detection of vibrations in buildings or workplaces [24,25].

As far as studies on vibrations in the human body are concerned, there are works on the study of vibrations during pregnancy [21], but much work remains to be completed, especially in neonatal patients, where a few studies have focused on the level of vibrations received during neonatal patient transport, both on land (ambulance) and by air (helicopter) [26,27,28,29,30] and this phenomenon as a cause of morbidity [26,29,31], but very little is known about the vibrations to which neonates are exposed during hospitalization [32].

The International Standard ISO 2631-2: 2003 [20] warns that the human response to vibration is very complex. On health damage caused by vibration, it adds that “biodynamic research studies as well as epidemiology have shown evidence of an elevated risk of health impairment due to prolonged exposure to high intensity whole-body vibration”. However, it also cites “there is not enough information to show a quantitative relationship between exposure to whole-body vibration in terms of probability of risk at various magnitudes and durations of exposure.” Spanish regulations on occupational diseases, the Royal Decree [33] which approves the list of occupational diseases in the Social Security System, mentions—among other diseases resulting from exposure to vibration—“Osteoarticular or neurovascular diseases caused by mechanical vibrations” or “dorsolumbar spine disorders caused by repetitive whole-body vibrations”. It is also known that being exposed to whole-body vibration generates acute health effects such as discomfort, interference with activities, alterations of physiological functions, neuromuscular, cardiovascular, endocrine and metabolic disturbances, and/or sensory disturbances of the central nervous system.

The equipment typically found in an NU room, such as alarms, pulse oximeters, infusion pumps, etc., has not been designed with the patients’ noise comfort in mind, nor has the room environment, such as central air conditioning. The geometric shape of the incubator clasp and the mechanism for attaching this type of alarm to the incubators amplify certain airborne noise frequencies, which was verified during the analysis of airborne noise in the neonatal ward of the hospital Fco. Borja in Gandía, Valencia, Spain [17]. This study demonstrates that there are airborne noise assessment scenarios where the level is higher inside the incubator than outside the incubator. This fact has been the source of the present study. It is necessary to determine if there is any path of transmission in the incubator itself that is fostering the propagation of certain frequencies that increase vibration levels. The frequency spectrum of the study must be broadened to assess what is happening at low frequencies. For this purpose, a protocolized study of the state of all existing vibrations in the neonatal ward must be carried out, and their transmission to the incubator studied in order to determine the level of vibration to which the newborns are exposed during their hospitalization. 

The main objective of this work is to study the vibrations to which neonatal patients are exposed during their hospitalization in a neonatal ward, focusing on those vibrations that reach the mattress where the neonate rests in the incubator. The study was conducted in the neonatal ward of the University Hospital Francesc de Borja in Gandía, Spain. 

In order to meet the objective, control measurements had to be performed in order to determine what kind of technical equipment should be used to record the measurements. For a correct vibration diagnosis, it must be determined what type of signal analysis to use, what type of accelerometers to use, the frequency spectrum where the study should be focused, as well as the recording times. It was also necessary to determine the possible study scenarios and situations in neonatal wards which could influence the vibrations that may reach the neonatal patient. With the recordings already made for vibrations and acceleration values per frequency, levels must be obtained and compared with legislation on permissible levels of vibration in hospital environments. 

## 2. Materials and Methods

### 2.1. Measuring Vibrations

There are different options for evaluating vibration values. A vibration is understood as a mechanical oscillation with respect to the point of equilibrium. This vibration, or displacement from the point of equilibrium, can be quantified in terms of displacement in meters (m), velocity in meters per second (m/s), or acceleration in meters per second squared (m/s^2^). The most commonly used variable in vibration assessment is acceleration. In this study, acceleration has been evaluated in m/s^2^, although all results will be expressed in decibels (dB). The dB is the most commonly used unit in acoustic engineering, all the control parameters that refer to acoustic quality are indicated in this way. A conversion to logarithmic scaling of the quantities must be considered, as expressed in Equation (1), where ‘*a*’ is the recorded acceleration in m/s^2^, *a*_0_ = 10^−6^ m/s^2^, which is known as reference acceleration, and *La* (dB) is the level of acceleration [24]: (1)$$La\;\left(\mathrm{dB}\right)=20\log\left(\frac{a}{a_{0}}\right)$$

The acceleration values in m/s^2^ recorded in the neonatal ward have been evaluated for each frequency value; however, some of the acoustic quality control parameters defined in the legislation [20,34] are referenced as global values. Therefore, it is necessary, in addition to the scale conversion defined in expression (1), to perform a third-octave weighting. In this weighting, the values are averaged per frequency in third-octave bands. Third-octave bands are fully defined, and each covers a different number of frequencies to be averaged, as well as having a different weight (w_m_) in the conversion [34]. Table 1 shows the central values of each third-octave band from 1 Hz to 80 Hz, the frequency range corresponding to each third-octave (upper and lower limit), as well as the weighting value for each band.

The vibration index *La_w_* is defined by the following expression:(2)$$La_{w}=20\log\left(\frac{a_{w}}{a_{0}}\right)\;\left(\mathrm{dB}\right)$$
where *a*_*w*_ is the value of the acceleration in each third-octave band, with the Wm factor, and is defined below (Equation (3)). This index is used in the estimation of maximum vibration values and annoyance during the assessment time and the interior space of buildings. In this study, we have used this index to evaluate the acoustic quality of the neonatal ward, although the focus of the study is on the neonatal incubator. This work considers the limits established by *La_w_* 37/2003 of 17 November on noise with regard to acoustic zoning, quality objectives, and acoustic emissions and Royal Decree 1367/2007 [35]. The publication of this legislation aims to prevent, monitor, and reduce the levels of vibro-acoustic pollution affecting the country. There are no specific limits for children. In [35] the following limit levels are proposed in order to achieve the European noise pollution mitigation target (Table 2).

As shown in Table 2, the regulation stipulates for the health sector that the recorded levels should be below the level of 72 dB, leaving a margin of 5 dB. The following procedure was used to obtain the vibration index [*La_w_* (dB)] from the accelerometer measurements of the acceleration per frequency and in m/s^2^: From the acceleration data in m/s^2^ and per frequency recorded on the accelerometer, the acceleration values in m/s^2^ in each third-octave are obtained, with a linear average from the upper/lower frequencies of each third, as indicated in the second and third columns of Table 1 (*a_j_*). In order to obtain the overall value, each of these values must be given the weight corresponding to the third octave to which they belong. The weights (*W_j_*) are defined in the fourth column of Table 1, and the value of each acceleration per third octave with its corresponding weight can be obtained from the product (*W_j_a_j_*). The sum of the value in all its octave-thirds expressed according to Equation (3) will give us the weighted acceleration *a_w_*, which will allow us, thanks to expression 2, to obtain the global acceleration index, *La_w_*.
(3)$$a_{w}=\sqrt{\underset{j}{\sum }{\left(W_{j}a_{j}\right)}^{2}}$$

### 2.2. Neonatal Ward, Study Scenarios

The neonatal ward is located on the first floor of the Francesc de Borja Hospital. This hospital is relatively new. It was opened in 2015 and provides medical care to 188,000 inhabitants. The main neonatal ward is rectangular in shape and has approximately 60 m^2^ of usable space. The number of cots and incubators in the room varies, with an average capacity of 6 incubators and a maximum of 10. It has a whole room air conditioning system. 

It is worth noting the large number of medical control devices installed in the ward, necessary for its proper functioning, many of which have audible alarms. In this work, results of the vibration analysis study are shown without any type of medical control device in operation, although the influence of the ward’s central air conditioning was assessed. 

Figure 1 shows images of the neonatal ward of the Francesc de Borja Hospital, as well as one of the neonatal incubators.

Records were made, with sensors distributed among the neonatal incubators, as shown in Figure 2. After a structural analysis of the neonatal incubator, 19 vibration positions are recorded in the incubator (Figure 2), thus trying to analyze all possible influences: the vibration transmission caused by the structure of the room itself, influence from the cabinet that all incubators have, influence of the motor (this motor controls the temperature, humidity, and oxygen conditions that the neonatal patient requires), influence of the mattress support plate, influence of the mattress, as well as the influence of the geometrical shape of the incubator canopy. All these recordings have been made in empty General Electric Health Care^®^ Giraffe incubators inside the NU. The recordings have been made on different days, time slots, and scenarios. The scenarios studied were: 

BGN: Background measurement, taken with the neonatal ward and incubator equipment turned off.

AirOn: Measured with the neonatal ward air conditioning turned on.

FAN_Motor: Measured with the incubator turned on.

FAN_Motor&AirOn: Measured with the incubator and neonatal ward air conditioning turned on.

All records were taken with a Brüel&Kjær 2250 sonometer in the advanced FFT analyzer module, using a Brüel&Kjær Type 4370 accelerometer. The first recordings were taken in the frequency range from 0 Hz to 500 Hz in order to detect possible anomalies at higher frequencies. For the vibration study, we focus on the frequency range of 0–80 Hz, as specified in Royal Decree 1367/2007 [35], which establishes the frequency range to be evaluated in the study of the Global Acceleration Level (*La_w_*). The value of the vibration index (dB) has been obtained, as indicated in [34].

## 3. Results

Figure 3 shows some of the accelerometer recording sites on the Giraffe incubator in the neonatal ward of the hospital.

A total of 456 records were made. The full measurement report is available in a Master’s thesis on Acoustical Engineering [36]. We will focus on those points of greatest relevance to the neonatal patient.

Results are shown for the measurement points: 11, 14, and 17. These points correspond to the inside of the incubator, in the bedding area, and at the level of the neonate’s head. Points 11, 14, and 17 are geometrically the same point but differ in the following: point 11 is just above the box where the incubator motor is located. On this surface, the incubator has a glass pane, point 14 being the recording on this pane. Finally, the mattress on which the newborn rests is placed on top of this glass, which is point 17. In addition, each point has been evaluated in different scenarios described above, namely: BGN (background noise measurement, taken with the neonatal ward equipment and incubator switched off), AirOn (measurement taken with the neonatal nursery air conditioner switched on), FAN_Motor (measurement taken with the incubator switched on) and FAN_Motor&AirOn (measurement taken with the incubator and neonatal ward air conditioner switched on).

Figure 4 shows an example of the first recordings that were made on site of the acceleration values (m/s^2^). The first recordings were made in the frequency spectrum from 0 Hz to 500 Hz. It is shown for point 11 and the different scenarios studied.

Figure 5 shows the average values of the recordings made, on different days, for the four scenarios studied in the frequency range 0–100 Hz. This frequency range is the frequency range necessary to obtain the acoustic quality parameters, as stated in [35]. The values are presented in dB, values of the acceleration level *La* (Equation (1)). Figure 5a shows the results for measurement position 11, Figure 5b for measurement position 14, and Figure 5c for measurement position 17.

Figure 6 shows the average values of the acceleration level *La* (dB), comparing the same FAN_Motor&Air scenario, points 11, 14, and 17: points on the motor (11), on the tray covering the motor (14), and on the mattress on top of the tray (17).

In order to be able to study in detail whether there is an influence on vibrations with the room air conditioning, Figure 7 shows the values of the FAN_Motor and FAN_Motor&Air scenarios at the same points shown above (points 11, 14, and 17). 

Figure 8 shows the overall vibration index (dB) of the points detailed in the previous charts, broken down by scenario.

It can be seen how for position 11, which is located above the motor structure, the levels obtained in the “Air” and “FAN_Motor&Air” scenarios exceed the levels indicated in the regulations and also exceed the maximum increase of 5 dB.

## 4. Discussion

This paper shows the results of the diagnosis of vibrations to which neonatal patients are exposed during their hospitalization in the neonatal ward. Levels have been recorded in different scenarios, and all have some vibrations. In some scenarios, global acceleration level values have been recorded that are higher than the values allowed by current legislation in a hospital environment.

In Figure 5a–c, the acceleration levels of three different points on the neonatal incubator are shown, and the influence of different scenarios such as air conditioning, incubator motor operation, or mattress are studied. It is observed that at the three points studied and in all the scenarios evaluated, there is a peak acceleration level between the bands of 16–20 Hz. This low frequency is fundamental for the analysis of the global value, as well as for the influence of vibrations on the human body. Therefore, we have found a starting point on which it will be necessary to act in order to reduce the vibration levels reaching the neonatal patient. Figure 6 shows the influence of the glass plate on the motor casing, as well as the influence of the mattress on the motor casing, reducing the acceleration levels. In Figure 7, in order to study the influence of the room air conditioning, the values of the frequency acceleration levels *La* (dB) of the scenarios with the incubator motor on (FAN_Motor) and with the incubator motor on the plus room air conditioning on (FAN_Motor&Air) have been compared. The results show that in the position over the motor casing, there is an influence of the air conditioning on the vibrations, but the influence of the air conditioning decreases when covering the motor casing with the glass plate, or even more with the mattress where the newborn rests. Figure 8 shows the comparisons of the global vibration index *La_w_* (dB), a detailed parameter that allows us to know if the legislated values for vibration levels are complied with within different uses of buildings, in the case in question, in hospital use. It is observed that at point 11, over the incubator motor casing, and in the scenario with the air conditioning and incubator motor turned on, the maximum permissible values are exceeded (77 dB) and that, with the air conditioning turned on in the room, it is very close to reaching the permissible value (72 dB). At points 14 and 17, the values are high but do not exceed the limit, which shows the importance of good damping, although not all vibrations are eliminated, and actually, we do not know if these low vibrations can affect neonatal health; this is the question to resolve in future studies.

The conclusion is that the greatest influence on vibrations is the incubator’s own motor, with other possible sources of vibrations, such as the room’s central air conditioning, having less influence.

In a recent review by Mc Calling [37], it is clear that incubators have the possibility of doing some harm as an unintended consequence of the design because they emit vibrations that can potentially have a negative impact on the newborn. ISO 2631 [20] details comprehensive methodologies for whole-body vibration measurement and describes a comfort rating scale to determine the severity of the exposure. Although legislation exists from an occupational perspective, there are currently no legal limits on neonatal exposure to vibration. There are very limited published data on vibration exposure in incubators during transport and even less “in situ” in the hospital environment. Most existing transport studies have limitations with respect to sample size, use of neonates versus the use of dummies, and modes of transport. However, the vibration emission data collected and published to date are at or above the upper end of the comfort scale classification according to ISO 2631. Studies [37] during in-hospital management to date also suggest that exposure to body vibrations in neonatal incubators exceeds the limit values in the vast majority of cases. This may worsen with the equipment of the unit and the alarms that this implies and also with the usual management of the neonate.

The point of discussion is whether this vibration is harmful (although any environmental pollution is supposed to be undesirable, especially for an immature organism).

The International Organization for Standardization has shown that vibrations have “adverse effects on the cardiorespiratory function, the central and peripheral nervous system, the electroencephalographic activity, body temperature, the metabolic and endocrine function and the gastrointestinal system” in adults. This Organization has established standards with respect to body vibration for seated adults. However, no standards have been adopted for people in other positions, such as standing, reclining, or lying down [25]. There is a standard for the permissible amount of noise that a baby in an incubator can safely endure, but there are no standards for the permissible amount of vibrations nor for their harmfulness.

Perhaps, in some cases, subjecting neonates to mild vibrations could be beneficial. According to the basic principle of stochastic resonance, the application of a modest amount of random vibration to a complex biological system, such as the human body, increases the sensitivity of that system. Stochastic resonance could be useful in preventing apnoea episodes in premature babies that are currently treated with caffeine or respiratory support systems.

A clinical trial showed that stochastic resonance stimulation using a special mattress could be used to treat premature babies with apnoea, bradycardia, and oxygen desaturation, with positive results. The mattress provided stimulation based on a gentle vibration similar to a soft massage. Stimulation showed promising results in three events, as apnoea was reduced by 50%, bradycardia intensity by 20%, and oxygen desaturation by 20% to 35% [38,39]. However, in this study, care was taken to avoid vibrations in the area corresponding to the baby’s head to avoid harmful effects on the still developing brain (unlike in our study where we have focused on the head resting area), and there are no more studies including the head’s area. 

## 5. Limitations

This study has only analyzed one type of incubator, so the results cannot be generalized to all types and brands, although it sets a precedent to be confirmed. It is necessary to do more studies in other neonatal units/hospitals to know different environmental or structural issues.

Vibrations should also be recorded when medical equipment is in operation because this can add some more vibration. 

This study has not considered possible variations in the position of the neonate within the incubator itself that could minimize the vibrations received.

## 6. Conclusions

This paper shows the results of the diagnosis of vibrations to which neonatal patients are exposed during their hospitalization in the neonatal ward of the Francesc de Borja Hospital in Gandía. The levels inside the incubator in the area where the neonate’s head rests and in different scenarios have been registered. These comparisons have indicated that, at least in the case study, all scenarios have some level of vibration that needs to be reviewed in more detail and solutions that need to be found in order to reduce the levels reaching the incubators in the neonatal wards. In some scenarios, global acceleration level values have been recorded that are higher than the values allowed by current legislation in a hospital environment. This study has highlighted the need to diagnose the state of vibrations to which neonatal patients are subjected during their admission. From this study, it is possible to protocolize the diagnosis of vibrations in any neonatal ward. Further studies are needed to understand the significance and impact of these findings in clinical practice.

## Figures and Tables

**Figure 1 healthcare-10-01180-f001:**
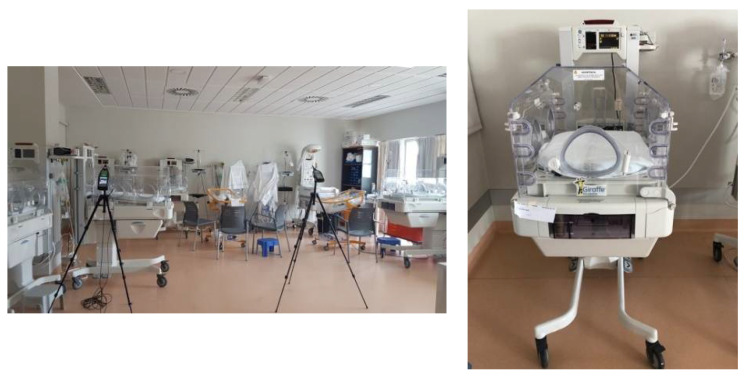
Neonatal ward of the Francesc de Borja Hospital in Gandía. Neonatal incubator.

**Figure 2 healthcare-10-01180-f002:**
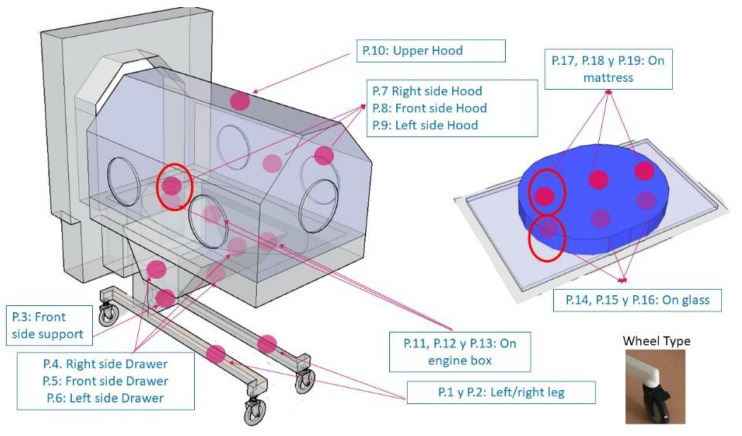
Positions of the recordings made in the incubator.

**Figure 3 healthcare-10-01180-f003:**
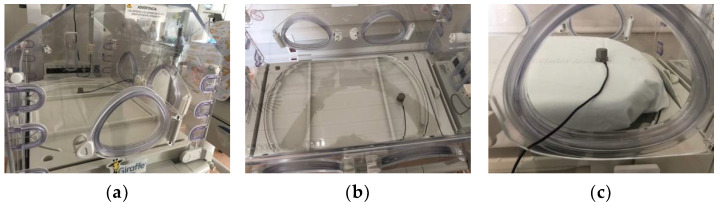
Examples of some of the recordings made on site in the neonatal incubator of the hospital. (**a**) Point 11 (correspond to the structure above the incubator motor). (**b**) Point 14 (correspond to the tray over the motor structure). (**c**) Point 17 (correspond to the mattress on the tray over the motor structure).

**Figure 4 healthcare-10-01180-f004:**
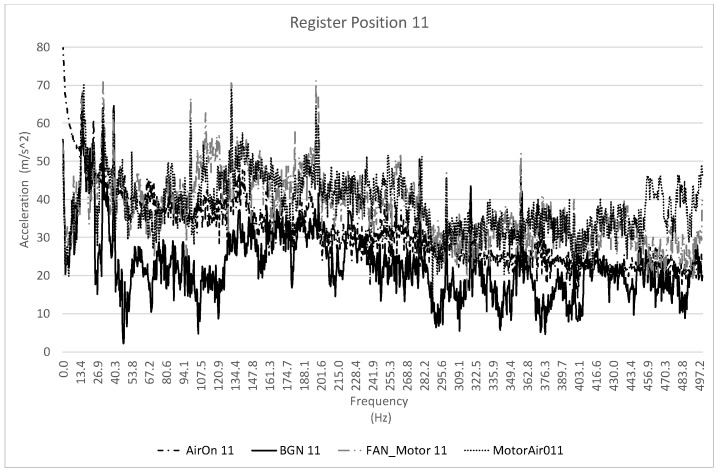
Example of one of the vibration recordings, in m/s^2^, at measurement position 11, for the four scenarios studied and in the frequency range 0–500 Hz.

**Figure 5 healthcare-10-01180-f005:**
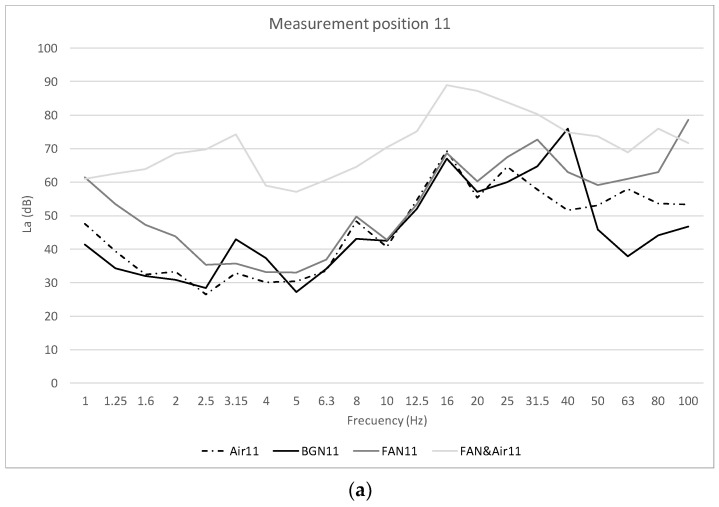
Acceleration levels *La* (dB), at measurement positions 11, 14 and 17, for the four scenarios studied and in the frequency range 0–100 Hz: (**a**) Position 11; (**b**) Position 14; (**c**) Position 17.

**Figure 6 healthcare-10-01180-f006:**
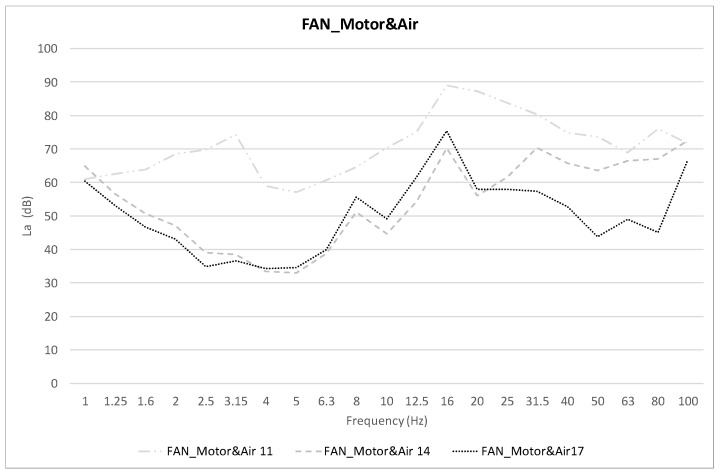
Average *La* (dB) levels. Positions 11, 14 and 17. FAN_Motor&Air Scenario.

**Figure 7 healthcare-10-01180-f007:**
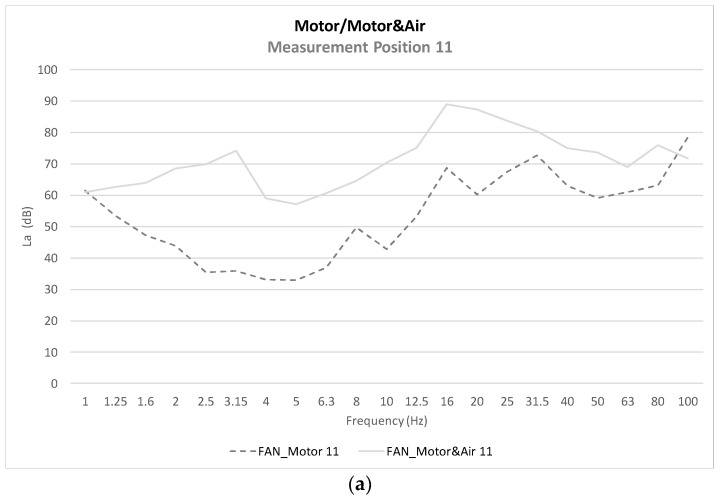
Average *La* (dB) levels comparative. FAN_Motor&Air Scenario. (**a**) Position 11; (**b**) Position 14; (**c**) Position 17.

**Figure 8 healthcare-10-01180-f008:**
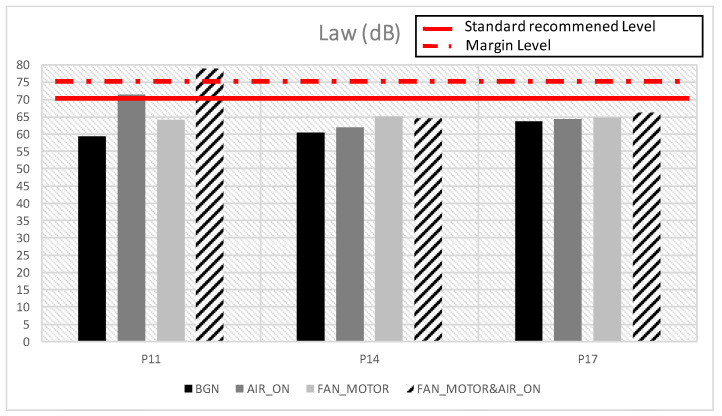
Average *La_w_* (dB) levels comparative. Positions 11, 14 and 17. FAN_Motor&Air Scenario.

**Table 1 healthcare-10-01180-t001:** Weighting values for third-octave bands center frequencies. Reference: weighting values Wm for third-octave bands center frequencies of 1 Hz to 80 Hz (ISO 2631-2:2011).

1/3 Octave Band Central Frequency (Hz)	Lower Frequency (Hz)	Higher Frequency (Hz)	$w_{m}\;Factor$
1	0.891	1.122	0.833
1.25	1.114	1.403	0.907
1.6	1.425	1.796	0.934
2	1.782	2.245	0.932
2.5	2.227	2.806	0.91
3.15	2.806	3.536	0.872
4	3.564	4.490	0.818
5	4.454	5.612	0.75
6.3	5.613	7.072	0.669
8	7.127	8.980	0.582
10	8.909	11.225	0.494
12.5	11.136	14.031	0.411
16	14.254	17.959	0.337
20	17.818	22.449	0.274
25	22.272	28.062	0.22
31.5	28.063	35.358	0.176
40	35.636	44.898	0.14
50	44.545	56.123	0.109
63	56.127	70.715	0.0834
80	71.272	89.797	0.0604

**Table 2 healthcare-10-01180-t002:** Noise quality objectives for vibrations applicable to the habitable interior space of buildings intended for residential, hospital, educational or cultural uses. Reference: Table C of Annex II. Noise quality objectives [35].

Use of the Building	Vibration Index, *La_w_* (dB)
Residential	75
Sanitary	72
Cultural or Educative	72

## Data Availability

Not available.

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
