# Peer review of "Vibroacoustic Study in the Neonatal Ward"

_healthcare, 2022, doi:10.3390/healthcare10071180_

Round 1
Reviewer 1 Report
Is the neonatal ward air conditioner, the air conditioning unit for the whole room or internal to each individual unit? This is not clear from the text and might be vital to understand the results and make recommendations for improvements.
Were the recordings at position 11 done without the glass plate and mattress or with both removed? The levels at 11 should be lower if the recordings are made with everything in place and the aid conditioner is external. Does it matter what the effect of the external air conditioner are on position 11 if they don’t make it to the infant lying on the mattress.
The introduction needs to introduce the problem of exposing neonates to vibrations. Even though there are few studies the reader should be presented the importance of the problem being addressed here. I would strongly recommend rewriting the introduction to discuss the health effects of exposure to vibrations in neonates. This part of the manuscript should focus on the particular contributions that this manuscript brings to current knowledge. A lot of the text is the discussion is repetitive and needs to be addressed in the introduction itself.
Abstract
P1. L 15-17. Consider rephrasing to make it easier to read. One possible way could be to say, “Due to the incubator’s motor position, the levels obtained in some scenarios measured in this study exceeded the regulation-specified levels.”
Introduction
P1. L35-37. Can you please cite the appropriate reference for this statement? The source (Environmental Commissioners from EU member states) seems very vague but is paraphrased.
P1. L40-42. Can you please rephrase this statement and make it easier to read? Is the frequency range for vibration defined based on health, well-being and perception? Can you please clarify if vibration depends on conduction through non-air media (directly coupled)?
P2. L51. Did you mean “detection/measurement” of vibrations instead of “diagnosis”?
P2. L71. Replace “has” with “have”. Consider rephrasing to something like “infusion pumps, central air conditioning etc., are not designed with the patient’s comfort/safety in mind.”
P2. L78-80. It is not clear if we are still focused on noise or vibration. You then say we need to look at vibration levels.
P2. L96. Its is not clear what these control indicators are. Are you referring to appropriate metrics to measure vibration?
P3. L113. It is not clear what these “acoustic quality control parameters” and “global” are in this context? Do you mean broadband\weighted sum when you say global there?
P4. Eqn 2. It is not clear what aw refers to here. Is the vibration index defined for each band?
P4. L128-129. Are either of these limits specific to infants? How should these limits change when applied to children?
P4. 136. What is the 5 dB margin for? Why use an arbitrary value of 5 dB? Why establish a safe-limit and then allow a random margin?
P4. L137. Consider replacing “It is considered appropriate to…” with “The following procedure was used to obtain the vibration index [Law (dB)] from the accelerometer…..”
P4. L145. Consider replacing “per third” with “in each third octave band”.
P5. L160. Consider replacing “was able to be” with “was”
P5. L164. Did you mean “Recordings were made, with sensors distributed among the neonatal incubators?
P5. L166. “analyze/analyse”
P5. L168. “This motor controls the temperature, humidity….”
P5. L183. “analyzer/analyser”
P5. Fig. 2 is it possible to highlight the three sensors for which the data is presented later?
P5. L185-P6. L186. Can you please clarify what these higher frequency anomalies are?
P6. L191. Consider saying, “Figure 3 shows some of the accelerometer recording sites on the Giraffe neonatal….”
P6. L197. Consider omitting “The extension of this report is beyond the scope of this paper,” The rest of that sentence can be left as is.
P6. L200-205. This could all be simplified to “The points 11,14 and 17, correspond to the structure above the incubator motor, the glass tray above the motor, on the mattress at the level of the neonate's head respectively.”
P7. Fig. 4. Should the y-axis not be in dB? The numbers look like they are in dB. You say twice above that the measurements are in dB. Consider presenting these in dB as well to keep things consistent.
P10. Fig 8. Check Spelling.
P11. L278. Can you please be more specific about the frequency at which the peak occurs? There seems to be a low value at 20 Hz.
P12. L 311-313. Is this claim based on your results or other results?
P12. L326-331. This material could be in the introduction.
P12. L329. Isn’t stochastic resonance imaging an imaging technique rather than a treatment?
Author Response
I am very grateful for your comments and i will do all changes required in order to improve the text. Thanks

Reviewer 2 Report
The manuscript reports on the amplitude of vibration in neonatal incubators. The manuscript includes background information about potentially-harmful effect of vibration. Previous reports seem to focus on adults in workplace settings, which aren't directly relevant to neonates. That does not make the general question unimportant or uninteresting, but a shorter review would be adequate to make the case that the study addressed an important issue. The text includes other information of marginal relevance; for example, see the Specific comment about Fig. 3 (Fig. 1 is also not very useful). The Discussion repeats some of the Results. Together, these choices make the manuscript seem too long for the limited results that it presents.
Specific comments:
lines 71-72: As written, this sentence says that central air conditioning is not designed for environmental comfort. I'm sure that's not what was intended; re-phrase to clarify what aspect of comfort is referred to.
line 145, "acceleration per third with": The word "octave" after "third" is missing.
Figure 3: This takes up much space but adds little information to what was shown in Fig 2; data from some of the locations illustrated are not even reported in the manuscript. Delete.
Figure 5: The labels include some English and some Spanish. The meaning is not in doubt, but it would be better to be consistent.
line 301-302: As written, this sentence seems to say that incubators are deliberately made to be potentially harmful. I believe the authors meant to say that the possibility of harm is an unintended consequence of the design. Re-phrase.
Author Response
I am very grateful for your comments and i will do all changes required in order to do a better text. Thanks.

Reviewer 3 Report
Vibrant topic, interesting subject!
I recommend a few modification.
Compare with study in neaby place.
Write short and clear conclusion usefull for all neonatal wards.
Double Check abrevioation.
Figure 6 maximize the writing.
Double check the English.
Author Response
I am very grateful for your comments and i will do all changes required in order to have a better text. Thanks

Round 2
Reviewer 2 Report
Most of my comments were addressed. Please check the x-axis labels of Figure 5 again for misspelled English words.